# Apical Sodium-Dependent Bile Acid Cotransporter, A Novel Transporter of Indocyanine Green, and Its Application in Drug Screening

**DOI:** 10.3390/ijms21062202

**Published:** 2020-03-23

**Authors:** Menq-Rong Wu, Jong-Kai Hsiao

**Affiliations:** 1Department of Medical Imaging, Taipei TzuChi General Hospital, Buddhist Tzu-Chi Medical Foundation, No.289, Jianguo Rd., Xindian Dist., New Taipei city 23142, Taiwan; nicole750707@gmail.com; 2School of Medicine, Tzu Chi University, No. 701, Sec. 3, Zhongyang Rd., Hualien 97004, Taiwan

**Keywords:** Apical sodium-dependent bile acid cotransporter, indocyanine green, drug screening

## Abstract

Bile acid plays critical roles in the elimination of inorganic compounds such as bilirubin, heavy metals, and drug metabolites. Apical sodium-dependent bile acid cotransporter (ASBT), a solute carrier membrane transport protein, transports bile acids. Several inhibitors of ASBT have been evaluated in clinical trials. Sodium taurocholate cotransporting polypeptide (NTCP), belonging to the same family as ASBT, has fluorescein 5(6)-isothiocyanate (FITC) and indocyanine green (ICG) transportability. ICG, a Food and Drug Administration-approved fluorophore at near-infrared range, has perfect optical characteristics, so it can be applied in cell tracking and drug screening. In this study, ASBT and NTCP were transduced into the HT-1080 cell line. Nude mice were subcutaneously xenografted with control and ASBT-expressing cells. ICG transportability was observed through flow cytometry, fluorescent microscopy, multi-mode plate readers, and an in vivo imaging system. Several molecules, including taurocholate, sodium deoxycholate, cyclosporine A, nifedipine, and Primovist, were used to evaluate an in vitro drug-screening platform by using the combination of ICG and ASBT through flow cytometry. ICG and FITC were validated and shown to be transported by ASBT. NTCP had a higher ICG intensity compared with ASBT. For cell tracking, the ASBT xenograft had similar ICG signals as the control. For a drug-screening platform, the ICG intensity decreased with 186 μM taurocholate (56.8%), deoxycholate (83.8%), and increased with nifedipine (133.2%). These findings are suggestive of opportunities for the high-throughput drug screening of cholestasis and other diseases that are related to the dynamics of bile acid reabsorption.

## 1. Introduction

Apical sodium-dependent bile acid cotransporter (ASBT), containing nine transmembrane domains, belongs to the solute carrier 10 (SLC10) family of membrane transport proteins [1]. Physically, ASBT transports the glycine and taurine conjugates of major bile acids, including cholic acid, deoxycholic acid, chenodeoxycholic acid, and ursodeoxycholic acid [2]. ASBT is significantly expressed on the apical side of enterocytes in the terminal ileum and is responsible for recycling bile acids by Na^+^-dependent symporting [3]. By contrast, sodium taurocholate cotransporting polypeptide (NTCP), also belonging to the SLC10 family, and organic-anion-transporting polypeptide 1B3 (OATP1B3), are expressed on the hepatic basolateral membrane responsible for recycling bile acid from the blood [1]. ASBT dysfunction results in more bile acids in the colon, leading to diarrhea, gallstone disease, hypertriglyceridemia, or even colon cancer [4]. However, the inhibition of ASBT, NTCP, or OATP1B3 could be a treatment option in hypercholesterolemia, because it prevents the intake of bile acids [5]. Currently, some inhibitors of ASBT have been discovered, such as calcium channel blockers (Nifedipine), statins, cyclosporine A, and SC-435 [1,4,6].

Indocyanine green (ICG), a Food and Drug Administration (FDA) approved drug (dye) for liver function testing and tumor detection, is a near-infrared fluorophore with better penetration and lower autofluorescent background compared with traditional fluorescent proteins, such as green fluorescent protein (GFP) and red fluorescent protein [7,8,9,10,11]. The influx and efflux of ICG occur through NTCP and OATP1B3 and multidrug resistance p-glycoprotein 3 (MDR3) and multidrug resistance p-glycoprotein 1 (MDR1), respectively [12,13,14]. In our previous study, we demonstrated the use of *NTCP* and *OATP1B3* as reporter genes combined with ICG for in vivo tumor cell tracking [15,16]. Moreover, the combination of NTCP and ICG can serve as a drug-screening platform in the treatment of hepatitis B and D virus (HBV/HDV) [16]. In terms of transporting molecules, ASBT, NTCP, and OATP1B3 have many similarities. Therefore, in this study, we hypothesized that ICG is the transporting molecule of ASBT.

Fluorescein 5(6)-isothiocyanate (FITC) is widely applied in the imaging field as a tracer. We validated that NTCP and OATP1B3 could transport FITC [17]. Moreover, the transporting molecule ASBT was highly crossed with NTCP and OATP1B3. Therefore, FITC was also considered to be transported through ASBT.

As more clinical trials on the regulation of bile acid in the elimination of heavy metal toxicity, reduction of cholestasis, and promotion of drug excretion are being conducted, ASBT inhibitors are one of the upcoming strategies [1]. Formerly, isotope-labeled taurocholate [4,18] was the only means for evaluating the drug–drug interaction of ASBT. However, ionizing radiation and isotope preparation decelerate the accessibility of the drug. Therefore, in this study, we aimed to establish a drug-screening platform for the ASBT inhibitor through optical characterization of the combination of ASBT transfected cell lines and ICG. 

## 2. Results

### 2.1. Confirmation of ASBT and NTCP Tranduction

In the beginning, we confirmed the ASBT and NTCP expression patterns in manipulated cells to ensure that ASBT could be functional. The C terminal of ASBT was fused with GFP, as well as NTCP. The ASBT and NTCP protein amounts were elevated, and the GFP signal was detected in cells expressing ASBT and NTCP in Western blotting (Figure 1a,d). The cellular location of ASBT and NTCP was visualized through the GFP signal (Figure 1b,e). Some of the ASBT was colocalized with actin, the membrane marker. The rest of the ASBT was located at the cytoplasm (Figure 1d). Most of NTCP was colocalized with actin (Figure 1f). In addition, the overexpression of ASBT and NTCP had no effect on proliferation (Appendix A).

### 2.2. ICG Intake through ASBT

#### 2.2.1. Confirmation of ICG Intake through ASBT

Cells expressing ASBT had a significantly increased ICG signal following treatment with 50 μg/mL ICG for 30 and 60 min, as observed using a multi-mode plate reader (Figure 2a). To further confirm ICG intake through ASBT, we validated it using different platforms. After treatment with 50 μg/mL ICG for 60 min, the increased ICG signal in cells expressing ASBT was validated through flow cytometry (Figure 2b) and confocal microscopy (Figure 2c).

#### 2.2.2. Pharmacokinetics of ICG Intake through ASBT

We have discussed the pharmacokinetics of ICG in cells expressing ASBT. The K_m_ of ICG in cells expressing ASBT and control cells was 12.63 and 100.9 mM, respectively (Figure 3a). Cells expressing ASBT had a greater ICG intensity, and difference in the decline of ICG was significant until 24 h after ICG treatment (Figure 3c). Moreover, ICG intake in cells expressing ASBT was significantly reduced upon cotreatment with taurocholate, which is a major transporting molecule through ASBT, compared with treatment with ICG (Figure 3b,d).

### 2.3. ICG Transportability between ASBT and NTCP

NTCP, of the same SLC10 family as ASBT, has ICG transportability [14]. We could get an idea about which of them could be a reporter gene by knowing the ICG transportability between ASBT and NTCP. The ICG transportability between ASBT and NTCP was compared. The in vivo imaging system (IVIS) showed that the detecting limitations of cells expressing ASBT and NTCP were 20,000 and 4000, respectively, after treatment with ICG (Figure 4a). ICG signals were further quantified at the cell number of 20,000. Only cells expressing NTCP had significantly increased ICG signals (Figure 4b). In flow cytometry, cells expressing ASBT and NTCP exhibited higher ICG signals compared with control cells (Figure 4c). Furthermore, cells expressing NTCP had higher ICG signals than those expressing ASBT. Cells expressing NTCP had a greater ICG intake efficiency. The ICG peak was reached following 10 min of ICG treatment in cells expressing NTCP, although the ICG signal gradually increased in cells expressing ASBT (Figure 4d).

### 2.4. FITC Intake through ASBT

A previous study found that NTCP and OATP1B3 transport FITC [16]. Moreover, the transporting molecule ASBT was highly crossed with NTCP and OATP1B3. Therefore, we testified that FITC was also considered to be transported through ASBT. After treating FITC for 5, 10, 30, and 60 min, cells expressing ASBT had a higher FITC signal than control cells (Figure 5a,c). The difference in FITC retained was distinguished from 0.5 to 6 h after the FITC treatment (Figure 5b,d). We further tested the FITC potential as an ASBT indicator in drug–drug interaction. ASBT-expressing cells with 186 μM taurocholate had a significantly lower FITC intensity than with 18.6μM taurocholate, suggesting ASBT as a FITC transporter. However, cells expressing ASBT had a higher FITC intensity in the cotreatment of FITC with taurocholate (Figure 5e), suggesting that FITC was not a suitable ASBT indicator.

### 2.5. Combination of ICG with ASBT for in Vivo Cell Tracking

Given that ASBT was the transporter of ICG, cells expressing ASBT were subcutaneously inoculated into SCID mice for cell tracking. ICG was administered on Day 13, 20, and 28 after xenografting, and the ICG signal was detected after one day of administration. From in vivo cell tracking, the ASBT-expressing tumor demonstrated similar ICG signals to the control tumor, and the GFP signal was much weaker compared with that in control cells (Figure 6a). The ASBT-expressing tumor had similar ICG signals to the control tumor, liver, and kidney in ex vivo imaging (Figure 6b). The ASBT-expressing tumor had greater ASBT positive staining compared with the control tumor (Figure 6c).

### 2.6. Combination of ICG with ASBT for Drug–drug Interaction

For drug screening, we chose bile salts (sodium taurocholate and sodium deoxycholate) and the ASBT inhibitors reported by Zheng et al. [4]. The ICG intensity was reduced on treatment with taurocholate in a dose-dependent manner in cells expressing ASBT. The ICG intensity with 186.0, 46.5, and 18.6 μM were 56.8%, 85.8%, and 98.7%, respectively. (Figure 7a). On treatment with deoxycholate, the ICG intensity, 83.8%, slightly reduced in cells expressing ASBT (Figure 7b). On treatment with cyclosporine A, the ICG intensity, 99.9%, appeared no different in cells expressing ASBT (Figure 7c). However, the ICG intensity, 133%, was slightly elevated on treatment with Nifedipine in cells expressing ASBT (Figure 7d). Furthermore, on treatment with Primovist (Gd-EOB-DTPA), the negative control, a similar ICG intensity (105.5%) to that in cells expressing ASBT treated with ICG (Figure 7e) was noted. To reduce the variation, the ICG intensity was further quantified and normalized with cells expressing ASBT treated with ICG. The data indicated significant differences in the ICG intensity on pretreatment with taurocholate and Nifedipine compared with no pretreatment (Figure 7f).

## 3. Discussion

In this study, we validated that ASBT transports fluorescent molecules, namely ICG and FITC. Moreover, we established an ASBT drug-screening platform that can be easily manipulated through flow cytometry with ICG as an indicator.

We used GFP-fused ASBT for exploring ICG’s intake ability through ASBT. ASBT and GFP looked diffused in Western blotting, which might be because of the glycosylation and GFP fusion (Figure 1a) [19]. ASBT is mainly expressed on the membrane [20]; however, the cellular location of ASBT was not only on the membrane, but also on the cytoplasm, which might have been due to the conjugation of GFP (Figure 1d). We still used GFP-fused ASBT for the study because some ASBT was still on the membrane, and could still be an in vitro model for exploration [18]. Previously, we successfully established an in vivo cell tracking model by using *NTCP* and *OATP1B3* as reporter genes combined with ICG [17]. In this study, *ASBT* was used as a reporter gene for in vivo cell tracking. However, we observed that distinguishing ASBT-expressing tumors from control tumors after ICG injection is not applicable. This may have been because the ICG intake efficiency was not adequately high in GFP-fused ASBT (K_m_ = 12.63 mM) in comparison to NTCP and OATP1B3. Moreover, we compared the ICG transport efficiency between NTCP and ASBT, and NTCP was observed to have a higher efficiency than ASBT in transporting ICG (Figure 4). Kramer et al. also revealed that NTCP has lower IC50 than ASBT when using ICG to compete with taurocholate (TA) [2]. Furthermore, *OATP1B3* as a reporter gene was better than *NTCP* [17]. Therefore, the reporter gene priority sequence was understood to be *OATP1B3*, *NTCP*, and lastly *ASBT* with the application of ICG.

FITC is widely used as a tracer in the imaging field. We discovered that NTCP and OATP1B3 could transport FITC [17]. In this study, we revealed that FITC was also transported by ASBT (Figure 5). Although ASBT, NTCP, and OATP1B3 could transport FITC, their application was limited. For instance, we attempted to address NTCP- and OATP1B3-expressing tumors with the FITC injection; however, the FITC signal was distributed everywhere [17]. In this study, the application of FITC as an indicator in ASBT drug–drug interaction (Figure 5e) was unsuccessful. The difficulty of FITC application may be because of its membrane diffusibility, low specificity (too many transporters could transport FITC), and interaction with the thiol group of FITC. 

The regulation of ASBT is now in the clinical therapy stage. ASBT dysfunction leads to more bile acids in the colon, leading to diarrhea, gallstone disease, hypertriglyceridemia, or even colon cancer [4]. However, the inhibition of ASBT, as well as NTCP, can be a treatment option for hypercholesterolemia and cholestasis, as well as nonalcoholic fatty liver [5,21,22]. Therefore, determining the drug–drug interaction of ASBT can help in medication choices. Herein, we used ICG as an indicator to establish a primary drug-screening platform through flow cytometry. Zheng et al. used isotope-labeled taurocholate as an indicator of drug screening [4]. Isotope-labeled taurocholate is highly sensitive and has a high transportability; however, the manipulation and synthesis of isotopes is time consuming and highly technically demanding. Because ICG is a near-infrared fluorescent dye, its detection is highly accessible through flow cytometry, high-throughput fluorescent microscopy, and a multi-mode plate reader. Although the ICG transportability was not high in ASBT (K_m_ = 12.63 mM) compared with taurocholate (K_m_ =209 μM) [23], this characteristic was used to select highly potent inhibitors of ASBT. Other molecules, such as Gd-B 20790 (92 ± 15 mM in OATP) [24], Primovist (4.1 mM in OATP1B3) [25], and glucose (~25 mM) [26] have a high K_m_ value. Moreover, the ICG intensity was reduced on the treatment of deoxycholate, transported through ASBT [27], which makes it more reasonable that makes ASBT was one of ICG transporters (Figure 7b). Therefore, we considered that ASBT was an ICG transporter. We would have liked to use specific ASBT inhibitors (S-8921, 2164U90, SC-435, and PR835) to perform the validation; however, these inhibitors were not easily accessible [28]. We chose other ASBT inhibitors (Nifedipine and cyclosporine A) and substrate (taurocholate and deoxycholate) for the validation. Nifedipine has been reported as an ASBT inhibitor; however, the ICG intensity increased on pretreatment with Nifedipine (Figure 7d,f), because Nifedipine is also an MDR1 inhibitor and the ICG efflux transporter [13,29]. The half maximal effective concentration (EC50) value of ICG for MDR1 is 15 ± 1.1 μM in Madin-Darby canine kidney (MDCK) cells with MDR1 overexpression [30]. The ICG intensity was significantly higher with Nifedipine than with cyclosporine A treatment in ASBT-expressing cells. However, their concentrations exhibited a huge difference; the Nifedipine and cyclosporine A concentrations were 288.7 and 2.1 μM, respectively. Therefore, at least two indicators are required for the drug-screening platform. Therefore, FITC was used as the second indicator; however, the FITC intensity increased on pretreatment with other molecules (Figure 5e). This was because FITC conjugates nonspecifically with other membrane molecules because it contains a thiol reacting group that can conjugate with other proteins [31]. 

The metabolism of bile acid includes the absorption of bile acid in the small intestine by means of ASBT and hepatic uptake by NTCP located at the basolateral membrane of the hepatocyte and OATP family at sinusoidal hepatocytes. In the current and previous studies, we discovered that ICG could be transported by OATP1B3, NTCP, and ASBT [15,16,17]. ICG has been proved as an in vivo imaging infrared fluorophore that has higher penetration. The use of ICG as an in vivo imaging tool for investigating bile acid metabolism for novel drug screening is feasible and could improve the process of drug development. 

There were some limitations in our study. The disrupted GFP-fused ASBT distribution might have affected the ICG transporting efficiency. Natural ASBT for evaluation will be addressed in a following study. The natural ASBT might have increased the ICG transportability and the difference in cell tracking and drug screening could have been more accessible. ICG transportability through ASBT did not verify the use of the ASBT knockout model. In our preliminary data, we gave ICG orally to the nude mice and tracked ICG signals using IVIS. All signals were in the digestive tract and none of them were in the periphery. We could harvest intestinal epithelial cells from ASBT knock-out mice to validate ICG transportability through ASBT in a further study. The number of molecules used to testify the drug-screening platform was low. The drug-screening platform needed a greater number of different molecules to make it possess confidence. 

In conclusion, ASBT was capable of transporting ICG, which is an FDA-approved infrared fluorescent dye that can be applied in a future drug-screening platform, in which potential ASBT-regulation medicine can be developed in a high-throughput and efficient manner that can have a clinical effect.

## 4. Materials and Methods 

### 4.1. Cell Culture and Preparation

HT-1080 cells were transduced with ASBT-, NTCP-, and GFP-carrying lentiviruses, followed by the Nature Protocols [32]. ASBT, NTCP, and GFP lentiviruses were made from RC210241L2 (Origene, Rockville, MD, USA), RC221202L2 (Origene, Rockville, MD, USA), and pCDH-EF1-MCS-BGH PGK-GFP-T2A-Puro (a generous gift from Dr. Yong-Chong Lin of National Taiwan University) vectors, respectively. OATP1B3-carrying HT-1080 cells were produced previously [15]. ASBT-expressing cells were derived from a single colony. NTCP-expressing cells were derived from sorting. These cells were cultured in minimum essential medium containing 10% fetal bovine solution (Biologic Industries, Cromwell, CT, USA), 100 U/mL penicillin, and 100 mg/mL streptomycin (Thermo Fisher Scientific, Waltham, MA, USA) in the incubator at 37 °C.

### 4.2. In Vitro ICG Intake

For the intake ability test, 2 × 10^4^ cells were seeded in 96 black wells for 1 day and treated with 50 μg/mL ICG for 0.25, 0.5, 1, 2, 3, 4, 5, 6, 10, 30, and 60 min. Subsequently, the cells were washed with 1 × phosphate buffer solution (PBS) thrice and detected using Spark 10M (Tecan Trading AG, Männedorf, Switzerland). 

Then, 1 × 10^6^ cells were seeded in a 6-well-plate for 1 day and treated with 50 μg/mL ICG for 1 h. Next, excess ICG was washed using 1× PBS. After the cells were trypsinized, the ICG intensity was detected using the FACSCalibur (BD Biosciences, San Jose, CA, USA) with an APC-Cy7 channel (785 nm) filter.

For the visualization of ICG, 2 × 10^4^ cells were seeded in 8-well-chamber slides for 1 day and treated with 50 μg/mL ICG for 1 h. Subsequently, the cells were washed with 1× PBS thrice and visualized using a TCS-SP5 laser-scanning microscope (Leica, Wetzlar, Germany) with a Cy5 filter.

For the comparison of ASBT and NTCP, 2 × 10^4^ cells were seeded in 96 black wells for 1 day and treated with 50 μg/mL ICG for 60 min. Subsequently, the cells were washed with 1 × phosphate buffer solution (PBS) thrice and detected using the IVIS Spectrum imaging system.

### 4.3. Pharmacokinetics of ICG

In a dose-dependent manner, 5 × 10^5^ cells were seeded in a 24-well-plate for 1 day and treated with 10, 50, 100, and 150 μg/mL ICG for 1 h. Next, excess ICG was washed using 1× PBS. After cells were trypsinized, the ICG intensity was detected using the FACSCalibur (BD Biosciences) with an APC-Cy7 channel (785 nm) filter. The value of K_m_ was based on the Michaelis–Menten model.

In a time-dependent manner, 5 × 10^5^ cells were seeded in a 24-well-plate for 1 day and incubated with 50 μg/mL ICG for 1 h. Subsequently, the cells were washed with 1× PBS thrice and detected using the FACSCalibur (BD Biosciences) with an APC-Cy7 channel (785 nm) filter at 0, 1, 4, 6, 24, and 48 h following ICG treatment.

### 4.4. The Animal Experiment of ICG Intake

Six to eight-week-old female BALB/cAnN.Cg-Foxnlnu/CrlNarl nude mice were purchased from the National Laboratory Animal Center. The animal study was approved by the Institutional Animal Use and Care Committee of Taipei Tzu Chi Hospital, Buddhist Tzu Chi Medical Foundation (106-IACUC-004, 3/7/2017). The mice were cared for based on the protocol of the Guide for the Care and Use of Laboratory Animals (National Institutes of Health).

Then, 1 × 10^6^ ASBT-carrying HT-1080 cells were subcutaneously administered to the right hind limb of nude mice. The left side was considered as the control. The tumor-bearing mice were intraperitoneally administered 10 mg/kg of ICG. The ICG signal was detected 1 d after ICG administration by using an IVIS Spectrum imaging system (Xenogen; Perkin Elmer) and traced on 14, 21, and 29 d after the xenograft. At 29 d after the xenograft, tumor-bearing mice were dissected to detect the ICG biodistribution using the IVIS Spectrum imaging system.

### 4.5. Drug-Screening Platform

A total of 5 × 10^5^ cells were seeded in a 24-well-plate for 1 day and pretreated with different molecules: 18.6, 46.5, and 186.0 μM sodium taurocholate (Millipore-Sigma, Billerica, MA, USA); 120.6 μM sodium deoxycholate (Millipore-Sigma); 2.1 μM cyclosporine A (Millipore-Sigma); 288.7 μM Nifedipine (Millipore-Sigma); and 1.25 mM Primovist (Gd-EOB-DTPA) (Bayer Pharma AG, Berlin, Germany) for 30 min. Subsequently, cells were cotreated with 12.9 μM ICG for 1 h more. The cells were then washed with 1× PBS thrice and detected using the FACSCalibur (BD Biosciences, San Jose, CA, USA) with an APC-Cy7 channel (785 nm) filter. All data were normalized with the number of cells expressing ASBT treated with ICG.

### 4.6. In Vitro Intake of FITC

For the intake ability test, 2 × 10^4^ cells were seeded in 96 black wells for 1 day and treated with 100 μM FITC (Millipore-Sigma) for 5, 10, 30, and 60 min. Subsequently, the cells were washed with 1× PBS thrice and detected using the Fluorescent Cell Imager (ZOE; Bio-Rad, Hercules, CA, USA).

For the declining test, 2 × 10^4^ cells were seeded in 96 black wells for 1 day and treated with 100 μM FITC for 60 min. Subsequently, the cells were washed with 1x PBS thrice and detected using the Fluorescent Cell Imager (ZOE; Bio-Rad, Hercules, CA, USA) at 0, 0.5, 1, and 3 h after FITC treatment.

All images were further analyzed using ImageJ for quantification. After acquiring the pixel number from GFP, the pixel number of the untreated sample was excluded from all data to reduce the noise from the background.

### 4.7. Cell Viability

A total of 1 × 10^3^ cells were seeded in a 96-well plate for 24, 48, and 72 h and treated with 0.5 mg/mL 3-(4,5-dimethylthiazol-2-yl]-2,5- diphenyltetrazolium bromide (MTT) for 2 h. Next, the MTT was removed and 100 μL of dimethyl sulfoxide was added to dissolve MTT. The signal was detected using Spark 10M (Tecan Trading AG, Männedorf, Switzerland).

### 4.8. Western Blotting

Primary antibodies against ASBT (1:1000; Thermo Fisher Scientific), NTCP (1:1000; Thermo Fisher Scientific), GFP (1:1000; TA180076; Origene, Rockville, MD, USA), and glyceraldehyde-3-phosphate dehydrogenase (GAPDH) (1:5000; Cell Signaling Technology, Danvers, MA, USA) were separated at 4 °C. Subsequently, the membrane was incubated with 1:5000 horseradish peroxidase-conjugated rabbit/mouse anti-IgG for 1 h at 25 °C.

### 4.9. Immunofluorescence and Immunohistochemistry

As the ASBT and NTCP plasmids contained GFP, GFP expression was directly observed using the Fluorescent Cell Imager (ZOE; Bio-Rad, Hercules, CA, USA) and co-stained with rhodamine phalloidin (Thermo Fisher Scientific) and DAPI. Tumors were fixed with 4% paraformaldehyde for paraffin sections. The 5-μm tissue slides were rehydrated for incubating them with anti-ASBT antibody (1:800; Thermo Fisher Scientific) overnight at 4 °C. Subsequently, the slides were incubated with an EnVision Kit (Agilent Technologies, Santa Clara, CA, USA) for 3, 3-diaminobenzidine staining and were counterstained with hematoxylin. All slides were observed using the ECLIPSE TE2000-U microscope (Nikon, New York, NY, USA).

### 4.10. Statistics

All experiments were performed in at least triplicate, and data were analyzed via one-way ANOVA and Newman–Keuls multiple comparison test using GraphPad Prism 5 software.

## Figures and Tables

**Figure 1 ijms-21-02202-f001:**
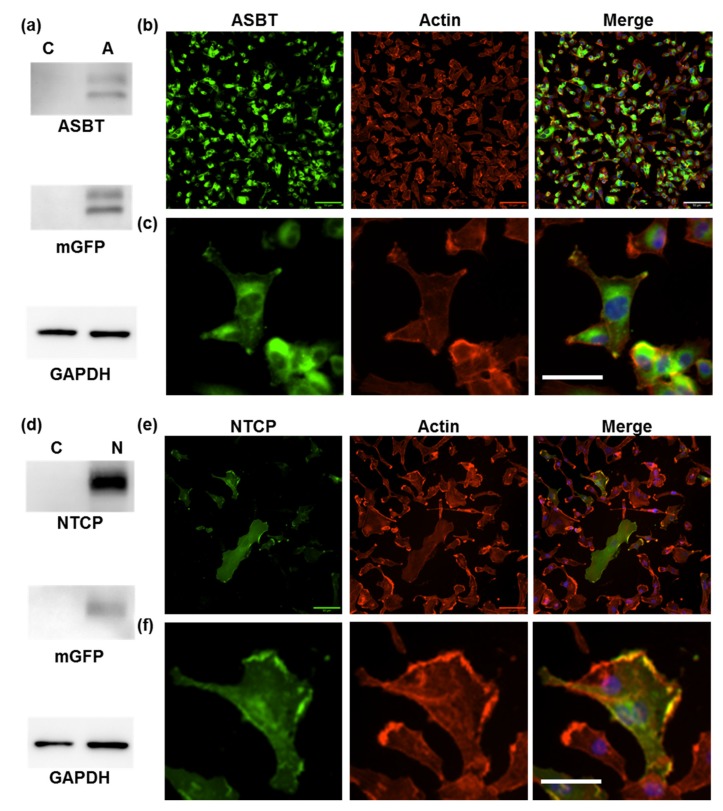
Overexpression of the apical sodium-dependent bile acid cotransporter (ASBT) and sodium taurocholate cotransporting polypeptide (NTCP). (**a**,**d**) Western blotting of ASBT, NTCP glyceraldehyde-3-phosphate dehydrogenase (GAPDH), and green fluorescent protein (GFP). (**b**,**e**) The location of ASBT, NTCP (green), actin (red), and nucleus (blue) in cells expressing ASBT at a lower magnification. Scale bar: 50 μm. (**c**,**f**) The location of ASBT, NTCP (green), actin (red), and nucleus (blue) in cells expressing ASBT at a higher magnification. Scale bar: 25 μm. C: control cells. A: cells expressing ASBT. N: cells expressing NTCP.

**Figure 2 ijms-21-02202-f002:**
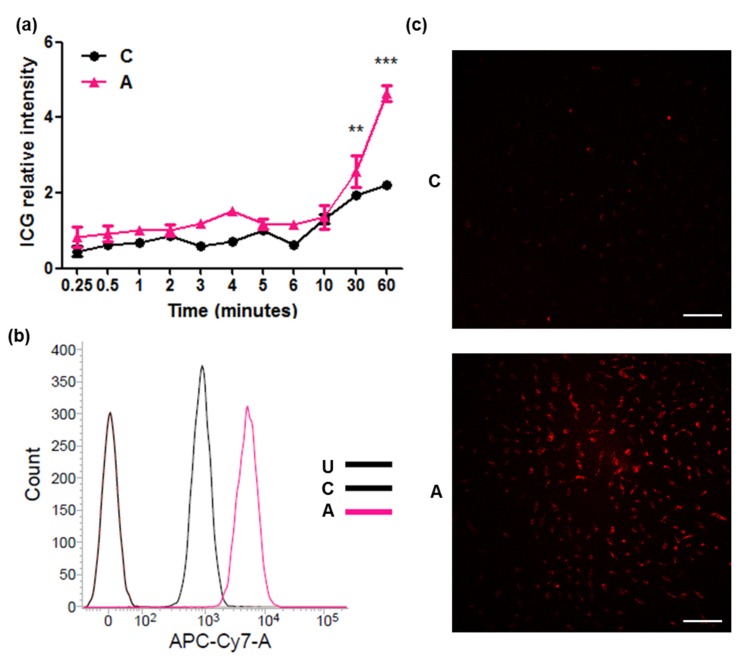
Indocyanine green (ICG) intake through ASBT. (**a**) The ICG intensity of treatment for different periods. (**b**) The ICG intensity detected through flow cytometry. (**c**) The ICG intensity visualized using confocal microscopy. C: control cells. A: cells expressing ASBT. U: untreated. Scale bar: 100 μm. Error bars show the SEM, where ** *p* < 0.01 and *** *p* < 0.001.

**Figure 3 ijms-21-02202-f003:**
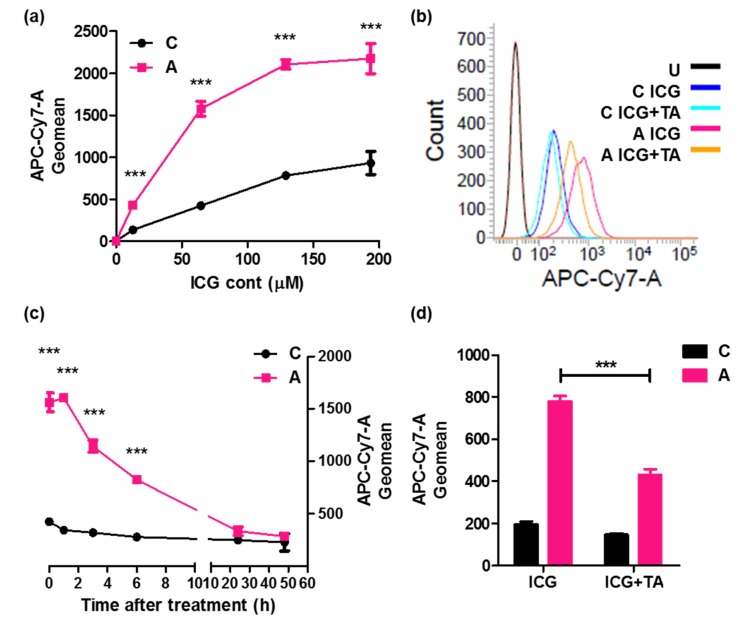
The pharmacokinetics of ICG in cells expressing ASBT. ICG intensity detected through flow cytometry (**a**) for treatment with different ICG doses, (**b**) after cotreatment with taurocholate, and (**c**) after ICG treatment to observe the decline. (**d**) The quantification from panel (B). C: control cells. A: cells expressing ASBT. U: untreated. TA: taurocholate. Error bars show the SEM, where *** *p* < 0.001.

**Figure 4 ijms-21-02202-f004:**
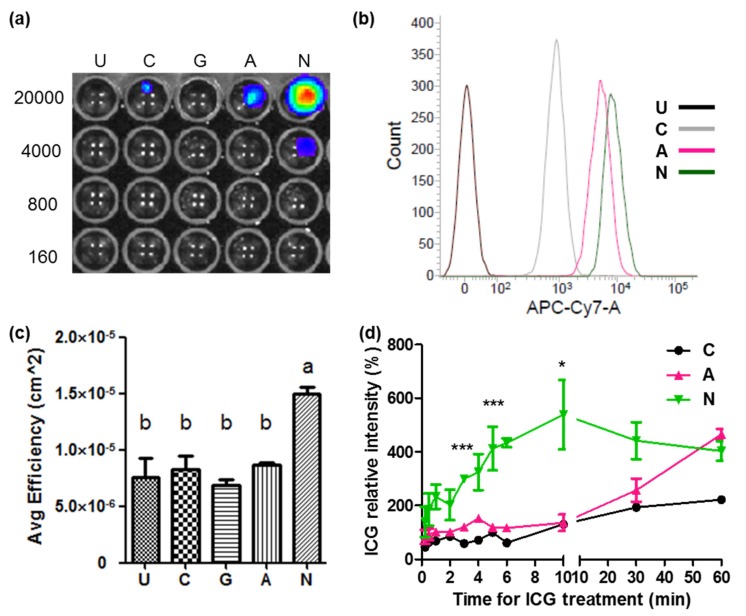
Comparison of ICG intake by ASBT and sodium taurocholate cotransporting polypeptide (NTCP). (**a**) The ICG intensity visualized using the in vivo imaging system (IVIS) at 20,000, 4000, 800, and 160 cell numbers. (**b**) The ICG intensity detected through flow cytometry. (**c**) Quantification of the ICG intensity with IVIS and analyzed using one-way ANOVA. The symbols a and b indicate the groups. (**d**) ICG intensity for different periods of treatment. The significant symbol was ASBT compared with NTCP. U: untreated. G: cells expressing GFP. C: control cells. A: cells expressing ASBT. N: cells expressing NTCP. Error bars depict the SEM, where * *p* < 0.05 and *** *p* < 0.001.

**Figure 5 ijms-21-02202-f005:**
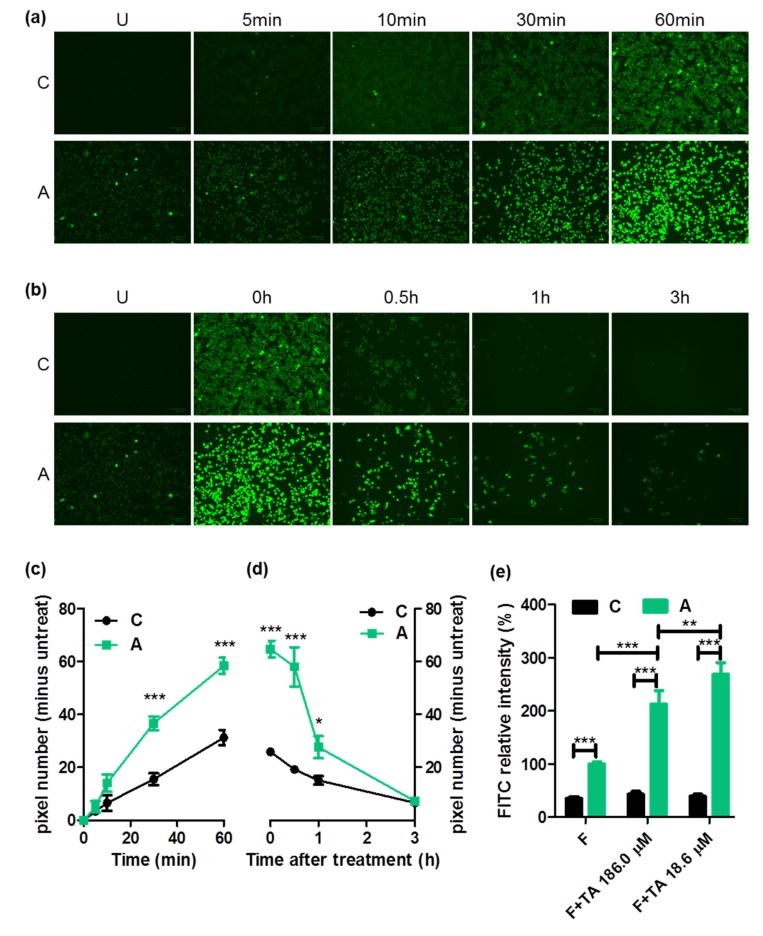
Fluorescein 5(6)-isothiocyanate (FITC) intake via ASBT. (**a**) FITC intensity visualized on treatment with FITC for 5, 10, 30, and 60 min. (**b**) Decline of FITC visualized at 0, 0.5, 1, and 3 h following FITC treatment. (**c**) The quantification from panel (a). (**d**) Quantification from panel (**b**). All quantification data were minus the background (untreated). (**e**) The completion of taurocholate (TA) and FITC using flow cytometry. C: control cells. A: cells expressing ASBT. U: untreated. Error bars denote the SEM, where * *p* < 0.05, ** *p* < 0.01, and *** *p* < 0.001.

**Figure 6 ijms-21-02202-f006:**
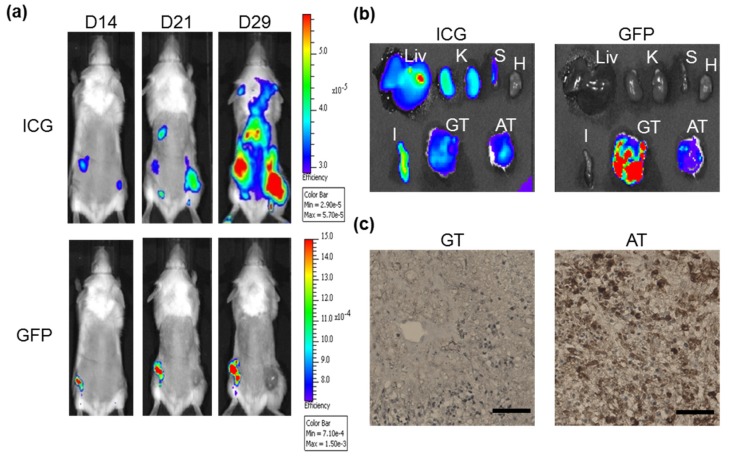
IVIS-based cell tracking with a combination of ICG and ASBT. (**a**) ICG and GFP signal were monitored at Day 14, 21, and 29 after xenografting. (**b**) The biodistribution of ICG and GFP. (**c**) ASBT expression identified by IHC. Liv: liver. K: kidney. S: spleen. H: heart. I: small intestine. GT: GFP-expressing tumor. AT: ASBT-expressing tumor. Scale bar: 160 μm.

**Figure 7 ijms-21-02202-f007:**
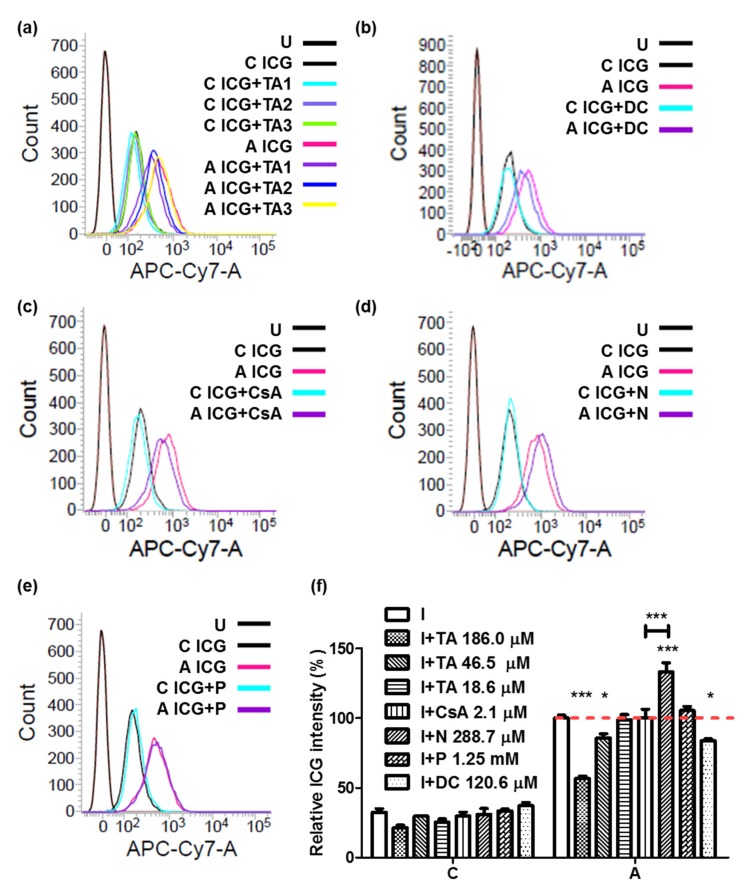
ICG-based drug-screening platform for ASBT through flow cytometry. (**a**) The selection of taurocholate (TA). TA1: 186.0 μM. TA2: 46.5 μM. TA3: 18.6 μM. (**b**) The selection of deoxycholate (DC), (**c**) cyclosporine A (CsA), (**d**) Nifedipine (N), and (**e**) Primovist (P). (**f**) The quantification of data from flow cytometry. C: control cells. A: cells expressing ASBT. U: untreated. Red dash line: the 100% ICG intensity. Error bars show the SEM, where * *p* < 0.05 and *** *p* < 0.001.

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
