# Peer review of "Apical Sodium-Dependent Bile Acid Cotransporter, A Novel Transporter of Indocyanine Green, and Its Application in Drug Screening"

_ijms, 2020, doi:10.3390/ijms21062202_

Round 1

Reviewer 1 Report

Abstract – the term toxic „materials” rather refers to inorganic compounds, should be changed

Introduction - multidrug resistance-associated protein 3 (MDR3) the term is not correct should be  multidrug resistance p-glycoprotein; and respectively multidrug resistance-associated protein 1 (MDR1) – multidrug resistance protein 1 (MDR1)

Results: Figure 1a – WB from transfected cells for ASBT and GFP are not conclusive, the picture quality and then the results are highly questionable; Figure 1d – high cytoplasmic but only weak membrane localization of ASBT is seen, which may affect functional study results (could it be related to C-terminus tagging).

Figure 3 and 4. The information about NTCP expression is not provided, and therefore conclusions on the transport capacity of NTCP and ASBT is not possible. The results simply reflect an experimental model conditions. Should not be generalized.

Figure 5. There is some evidence that FITC was transported by ASBT, at least co-administration with taurocholate should be evaluated.

Figure 1S. For ASBT specificity an interaction with taurocholate should be assessed.

Discussion. The statements that ASBT was capable of transporting ICG, and then can be applied as drug screening platform is not univocally documented, as in vivo experiments provide even contrary findings. Also FITC due to its complex transmembrane transport is also not a suitable marker of ASBT transporter activity. The discussion should clearly reflect these limitations.

Author Response

Abstract – the term toxic „materials” rather refers to inorganic compounds, should be changed

Ans:

Thank you for your notification. We have changed “toxic material” into “ inorganic compounds”.

Introduction - multidrug resistance-associated protein 3 (MDR3) the term is not correct should be  multidrug resistance p-glycoprotein; and respectively multidrug resistance-associated protein 1 (MDR1) – multidrug resistance protein 1 (MDR1)

Ans:

Thank you for your remind. We have corrected these terms.

Results: Figure 1a – WB from transfected cells for ASBT and GFP are not conclusive, the picture quality and then the results are highly questionable; Figure 1d – high cytoplasmic but only weak membrane localization of ASBT is seen, which may affect functional study results (could it be related to C-terminus tagging).

Ans:

We performed western blotting again and renewed our Figure 1a. There were two bands that revealed both membranes with ASBT and GFP antibodies hybridization.

We still used GFP-fused ASBT for the study because some ASBT was still on the membrane, and could even be an in vitro model for exploration. The disrupted GFP-fused ASBT distribution might affect the ICG transporting efficiency. Moreover, natural ASBT rather than GFP-fused ASBT will be carried out in our following study.  We discussed this issue in the second paragraph of the discussion panel)

Figure 3 and 4. The information about NTCP expression is not provided, and therefore conclusions on the transport capacity of NTCP and ASBT is not possible. The results simply reflect an experimental model conditions. Should not be generalized.

Ans:

Thank you for your comment. We provided data in the new figure 1d~f.

Figure 5. There is some evidence that FITC was transported by ASBT, at least co-administration with taurocholate should be evaluated.

Ans:

ASBT expressing cell with 186 μM taurocholate had significantly a lower FITC intensity than with 18.6μM taurocholate, suggesting ASBT as one of the  FITC transporter. However, cells expressing ASBT had a higher FITC intensity in the cotreatment of FITC with taurocholate (new Figure 5e). It might be the interaction of the thiol group of FITC with taurocholate.

We found the difficulty of FITC application because of its membrane diffusibility, low specificity (too many transporters could transport FITC), and interaction with the thiol group of FITC. We discussed this issue in the third paragraph of the Discussion panel.

Figure 1S. For ASBT specificity an interaction with taurocholate should be assessed.

Ans:

Thank you for your valuable comments. Taurocholate, a nature bile acid, was transported through ASBT. Both Sun et al. (J. Biol. Chem. 2006, 281, 16410–16418.) and Zheng et al. (Mol. Pharm. 2009, 6, 1591–1603.) use taurocholate to address the drug-drug interaction of ASBT. That is why we chose taurocholate as a competitor to against ICG. We added these references in the introduction panel and also the results we did has been added to the results panel (Pharmacokinetics of ICG intake through ASBT) and Figure 3b and 3d.

Discussion. The statements that ASBT was capable of transporting ICG, and then can be applied as drug screening platform is not univocally documented, as in vivo experiments provide even contrary findings. Also FITC due to its complex transmembrane transport is also not a suitable marker of ASBT transporter activity. The discussion should clearly reflect these limitations.

Ans:

Thank you for your suggestion. We addressed the limitation in the 6th paragraph of the Discussion panel as follows:  The disrupted GFP-fused ASBT distribution might affect the ICG transporting efficiency. Natural ASBT for evaluation will be addressed in the following study. The natural ASBT might increase the ICG transportability, and the difference in cell tracking and drug screening could be more accessible. ICG transportability through ASBT did not verify the use of the ASBT knock-out model. We could harvest intestinal epithelial cells from ASBT knock-out mice to validate ICG transportability through ASBT in a further study. All signals were in the digestive tract and none of them were in the periphery. The number of molecules used to testify the drug screening platform was low. The drug screening platform needs a greater number of different molecules to make it possess confidence.

Reviewer 2 Report

The authors have reported that ICG is an ASBT substrate which could be used as a diagnostic agent. However, there are some concerns associated with the study that needs detailed explanation as listed below:

  1. The abstract is not covering all the aspects of the study. The author should represent some quantitative data in the abstract to give a clear perspective to the readers. In the present form, it is too generalized.
  2. The Km value of ICG for ASBT is in millimolar level which questions on its specificity. Please clarify. Additionally, please add the information about any substrates that has been reported as a specific substrate for any transporter at such high Km value.
  3. The authors have used nifedipine that inhibited both ASBT and MDR1, however, accumulation of ICG was observed within the cells indicating that ICG is being taken up by the cells through other transporters or passive diffusion and not a specific substrate for ASBT.
  4. Was the difference between the ICG uptake in the presence of Cyclosporine (specific MDR1 inhibitor) and Nifedipine (ASBT and MDR1 inhibitor) statistically significant?
  5. What is the IC50 value of ICG for MDR1? Please clarify.
  6. Wouldn't it have been a straightforward experiment if the authors would have used ASBT knock-out mice and then dosed the animals with ICG orally and measured the portal vein concentration following the ICG uptake via ASBT in jejunum and ileum and compared the data with wild-type animals.
  7. Based on the data presented, it could only be concluded that ICG is a substrate for ASBT with no specificity considering high Km value and being taken up by the cells in the presence of ASBT inhibitor.

Author Response

The authors have reported that ICG is an ASBT substrate which could be used as a diagnostic agent. However, there are some concerns associated with the study that needs detailed explanation as listed below:

  1. The abstract is not covering all the aspects of the study. The author should represent some quantitative data in the abstract to give a clear perspective to the readers. In the present form, it is too generalized.

Ans:

Thank you for your valuable opinions. We rewrote the abstract to cover all the aspects of the study. We also provided valuable quantitative date in the abstract.

  1. The Km value of ICG for ASBT is in millimolar level which questions on its specificity. Please clarify. Additionally, please add the information about any substrates that has been reported as a specific substrate for any transporter at such high Km value.

Ans:

Thank you for your suggestion. We made the calculation based on the data from figure 3a. We understand that the Km value of ICG for ASBT was quite high. Other molecules have a high Km value, such as Gd-B 20790 (92 ± 15 mM in OATP), Primovist (4.1 mM in OATP1B3), and glucose (~25 mM). Therefore, we considered that ASBT was an ICG transporter. Moreover, the ICG intensity was reduced on the treatment of deoxycholate, transported through ASBT, that makes ASBT was one of ICG transporters more reasonable. This issue has been added to the fourth paragraph of the Discussion panel(new Figure 7b).

  1. The authors have used nifedipine that inhibited both ASBT and MDR1, however, accumulation of ICG was observed within the cells indicating that ICG is being taken up by the cells through other transporters or passive diffusion and not a specific substrate for ASBT.

Ans:

Thank you for your comments. NTCP and OATP1B3 are also transporting ICG. That is why we overexpressed ASBT and compared it with mock control to reduce other interference.

  1. Was the difference between the ICG uptake in the presence of Cyclosporine (specific MDR1 inhibitor) and Nifedipine (ASBT and MDR1 inhibitor) statistically significant?

Ans:

Thank you for your important question. ICG intensity was significantly higher after nifedipine treatment than cyclosporine treatment in ASBT expressing cells. However, their concentrations exhibited a huge difference; the nifedipine and cyclosporine A concentration was 288.7 and 2.1 μM, respectively. We discussed this issue in the fourth paragraph of the Discussion panel.

  1. What is the IC50 value of ICG for MDR1? Please clarify.

Ans:

Thank you for your suggestion. We think it will be more appropriate to use EC50 since MDR1 is responded to the ICG efflux. The half-maximal effective concentration (EC50) value of ICG for MDR1 is 15 ± 1.1 μM in MDCK-MDR1 cells. This critical issue has also been listed in the fourth paragraph of the Discussion panel.

  1. Wouldn't it have been a straightforward experiment if the authors would have used ASBT knock-out mice and then dosed the animals with ICG orally and measured the portal vein concentration following the ICG uptake via ASBT in jejunum and ileum and compared the data with wild-type animals.

Ans:

Thank you for your suggestion. We would like to perform what you suggested. However, we noticed one published result mentioning the intestine doesn’t absorb ICG although ASBT expression in jejunum and ileum (Ref: Gut, 1976, 17, 588-594). Besides, we injected ICG to nude mice through oral administration and observed the ICG signal by IVIS. All signals were in the digestive tract, and none of them in the periphery. Consequently, we will do the ASBT knock-out mice model in a new strategy in the following study, and This issue has been discussed in The sixth paragraph of the Discussion panel.

  1. Based on the data presented, it could only be concluded that ICG is a substrate for ASBT with no specificity considering high Km value and being taken up by the cells in the presence of ASBT inhibitor.

Ans:

Thank you for your comments. We understand that the Km value of ICG for ASBT was quite high. However, some of the transporter for similar mechanisms also share high Km such as Gd-B 20790 (92 ± 15 mM in OATP), Primovist (4.1 mM in OATP1B3), and glucose (~25 mM). Therefore, we considered that ASBT was an ICG transporter. Moreover, the ICG intensity was reduced on the treatment of sodium deoxycholate, transported through ASBT, that makes ASBT was one of ICG transporters more reasonable (new Figure 7b). This critical issue has been discussed in the fourth paragraph of the discussion panel.

The disrupted GFP-fused ASBT distribution might affect the ICG transporting efficiency. We still used GFP-fused ASBT for the study because some ASBT was still on the membrane, and could still be an in vitro model for exploration. This might be a reasonable cause a high Km value. Natural ASBT for evaluation will be addressed in the following study. This issue has also been listed in the second paragraph of the Discussion panel.

Round 2

Reviewer 1 Report

I would suggest to accept the manuscript in its present form.

Reviewer 2 Report

The authors have answered the comments satisfactorily.